# Unmet need of essential treatments for critical illness in Malawi

**Raphael Kazidule Kayambankadzanja**[1,2]*, **Carl Otto Schell**[3,4,5], **Isaac Mbingwani**[1,6], **Samson Kwazizira Mndolo**[2], **Markus Castegren**[7,8], **Tim Baker**[1,2,3,8]

**1** University of Malawi, College of Medicine, Blantyre, Malawi, **2** Department of Anaesthesia and Intensive Care, Queen Elizabeth Central Hospital, Blantyre, Malawi, **3** Health Systems & Policy, Department of Global Public Health, Karolinska Institutet, Stockholm, Sweden, **4** Centre for Clinical Research Sörmland, Uppsala University, Eskilstuna, Sweden, **5** Department of Internal Medicine, Nyköping Hospital, Sörmland Region, Nyköping, Sweden, **6** Chiradzulu District Hospital, Chiradzulu, Malawi, **7** CLINTEC and FyFa, Karolinska Institutet, Stockholm, Sweden, **8** Perioperative Medicine & Intensive Care, Karolinska University Hospital, Stockholm, Sweden

* raphkazidule@gmail.com

## Abstract

### Background

Critical illness is common throughout the world and has been the focus of a dramatic increase in attention during the COVID-19 pandemic. Severely deranged vital signs such as hypoxia, hypotension and low conscious level can identify critical illness. These vital signs are simple to check and treatments that aim to correct derangements are established, basic and low-cost. The aim of the study was to estimate the unmet need of such essential treatments for severely deranged vital signs in all adults admitted to hospitals in Malawi.

### Methods

We conducted a point prevalence cross-sectional study of adult hospitalized patients in Malawi. All in-patients aged $\geq$18 on single days Queen Elizabeth Central Hospital (QECH) and Chiradzulu District Hospital (CDH) were screened. Patients with hypoxia (oxygen saturation <90%), hypotension (systolic blood pressure <90mmHg) and reduced conscious level (Glasgow Coma Scale <9) were included in the study. The a-priori defined essential treatments were oxygen therapy for hypoxia, intravenous fluid for hypotension and an action to protect the airway for reduced consciousness (placing the patient in the lateral position, insertion of an oro-pharyngeal airway or endo-tracheal tube or manual airway protection).

### Results

Of the 1135 hospital in-patients screened, 45 (4.0%) had hypoxia, 103 (9.1%) had hypotension, and 17 (1.5%) had a reduced conscious level. Of those with hypoxia, 40 were not receiving oxygen (88.9%). Of those with hypotension, 94 were not receiving intravenous fluids (91.3%). Of those with a reduced conscious level, nine were not receiving an action to protect the airway (53.0%).

**Data Availability Statement:** Due to the sensitive nature of the information about severely unwell patients, and the requirements of the College of Medicine Research and Ethics Committee, data

from the study are not publicly available. Data can be requested by contacting the corresponding author raphkazidule@gmail.com or the College of Medicine Research and Ethics Committee comrec@medcol.mw including a motivation for the request for the data and a detailed study plan.

**Funding:** TB received grants for the study from the Swedish Research Council, Laerdal Foundation, the Association of Anaesthetists of Great Britain and Ireland, Martin Rinds Stiftelse and Life Support Foundation. The funders had no role in study design, data collection and analysis, decision to publish, or preparation of the manuscript.

**Competing interests:** The authors have declared that no competing interests exist.

## Conclusion

There was a large unmet need of essential treatments for critical illness in two hospitals in Malawi.

## Introduction

Critical illness is common throughout the world [1] and has been the focus of a dramatic increase in attention in the COVID-19 pandemic [2]. Severely deranged vital signs can identify critical illness, are simple to check [3] and treatments that aim to correct derangements are established, basic and low-cost [4]. Previously, we have argued that as such treatments could improve outcomes they should be part of Universal Health Coverage, [5] and they should not be forgotten in efforts to tackle the COVID-19 pandemic [6]. The coverage, (and its reverse, unmet need), of these treatments in hospitals in low-income countries (LICs) and elsewhere is not known. The aim of the study was to estimate the unmet need of essential treatments for severely deranged vital signs in all adults admitted to hospitals in Malawi. In this study, we use three selected vital signs: oxygen saturation (hypoxia), blood pressure (hypotension), and low Glasgow Coma Scale. These vital signs were chosen for their simplicity and the cut offs were chosen as they have previously shown to be markers of danger signs in critically ill patients.

## Methods

We conducted a point prevalence cross-sectional study with follow-up of adult hospitalized patients in Malawi. A team of nurses and senior nursing students screened all in-patients aged ≥18 on single days in January 2017 and May 2018 in Queen Elizabeth Central Hospital (QECH) and November 2017, February 2018 and July 2018 in Chiradzulu District Hospital (CDH). CDH had three data collections to increase representation from the facility. We included all adult patients in the hospitals. Data on patient demographics, diagnoses and prescribed treatments were collected from the patients' files and clinical data on the patients' vital signs and ongoing therapies were collected by direct clinical observation at the time of inclusion. We used Omron M2 and Lifebox pulse oximeters to measure blood pressure and oxygen respectively. Consciousness was measured using the Glasgow Coma Scale (GCS). Patients were considered to have an altered mental status if they had a GCS score of less than 9. Data were collected by qualified nurses and senior nursing students following training that was conducted the day before to ensure a standardised methodology.

QECH is a 1500 bedded large, referral hospital with four adult ICU beds that provide mechanical ventilation, vasopressors and close monitoring. CDH is a 300 bedded district hospital without an ICU. These two hospitals were chosen to provide settings of different resources, staffing and epidemiology. Both are public facilities that provide acute care free of charge to all patients. QECH is a referral centre for CDH and other district hospitals. None of the patients had an advanced directive precluding administration of the treatments during the study period. Patients with hypoxia (oxygen saturation <90%), hypotension (systolic blood pressure <90mmHg) and reduced conscious level (GCS <9) were included in the study. The cut-offs for severe vital sign derangements were adopted from previous work in Tanzania and Sweden [4,7]. These parameters were chosen for their simplicity and for measuring target treatments.

The a-priori defined essential treatments were oxygen therapy for hypoxia, intravenous fluid for hypotension and an action to protect the airway for reduced consciousness (placing the patient in the lateral position, insertion of an oro-pharyngeal airway or endo-tracheal tube or manual airway protection). These essential treatments were regarded as being present if the patient was observed to be receiving the treatment at the time of inclusion into the study. The treatments were chosen as they are regarded as standard medical practice and were expected to be available in all wards in both the study hospitals. Data on admitting specialty, previous surgery and HIV-status were retrieved from the medical records. Stata (Release 15, StataCorp, College Station, TX) was used for analysis. The study followed established ethical principles, written, informed consent was obtained from all participants and for those with altered mentation, the participant's next-of-kin provided the informed consent. The on-duty ward in-charge was immediately informed whenever a patient was found with a severely deranged vital sign. Ethical approval was granted by the University of Malawi College of Medicine Research and Ethics Committee (COMREC P.08/16/2007).

## Results

Of the 1135 hospital in-patients screened, 45 (4.0%) had hypoxia, 103 (9.1%) had hypotension, and 17 (1.5%) had a reduced conscious level (Table 1) The median (IQR) age of patients with a severely deranged vital sign was 39.5 years (30–54). Characteristics of all screened patients are in the S1 Table. Among the screened patients 5 were in ICU and had no unmet need.

Of those with hypoxia, 40 were not receiving oxygen (88.9%). Of those with hypotension, 94 were not receiving intravenous fluids (91.3%). Of those with a reduced conscious level, nine were not receiving an action to protect the airway (53.0%).

Ten of the hypoxic patients (22.2%; 95% CI 0.11–0.37), 23 of the hypotensive patients (22.3%, 95% CI; 0.15–0.32) and nine of the patients with a reduced conscious level (53.0%, 95% CI; 0.28–0.77) died in-hospital. The deaths occurred after a median of 2.6 days (1.2–7.4).

**Table 1. Participant characteristics, in-hospital mortality rates and the unmet need of essential treatments, by deranged vital sign.**

| All n(%) unless stated | Hypoxia (oxygen saturation <90%) | Hypotension (systolic blood pressure <90mmHg) | Reduced conscious level (Glasgow Coma Score <9) |
|---|---|---|---|
| N | 45 | 103 | 17 |
| Age median (Inter Quartile Range) | 58 (31–65) | 36 (29–48) | 49 (32–62) |
| Sex (Female) | 25 (55.6%) | 62 (60.2%) | 7 (41.2%) |
| HIV positive / HIV status known (%) | 12/31 (38.7%) | 62/87 (71.3%) | 8/12(66.7%) |
| Had surgery in hospital | 1 (2.2%) | 11 (10.7%) | 4 (23.5%) |
| Admitting specialty | | | |
| Medicine | 37(82.2%) | 80 (77.7%) | 10 (58.8%) |
| Surgery | 8 (17.8%) | 18 (17.5%) | 7 (41.2%) |
| Obstetrics and Gynaecology | 0 | 5 (4.9%) | 0 |
| In-hospital mortality | 10 (22.2%) | 23 (22.3%) | 9 (53.0%) |
| Unmet need of essential treatment * | 40 (88.9%) | 94 (91.3%) | 9 (53.0%) |

* unmet need refers to the number and proportion of patients with a deranged vital sign who were not receiving the needed, corresponding, essential treatment.

## Discussion

We have found a large unmet need of essential treatments for critical illness among unselected adult in-patients in Malawi. Unmet need has been studied in other fields [8] and is useful for evaluating the processes of care and to highlight areas for potential quality improvement. To our knowledge, the unmet need and the hospital-wide quality of care in critical illness have not previously been studied in a low-income country. Our findings are striking, given that the treatments were chosen due to their simplicity, low-cost and availability in the study hospitals. We chose three single vital signs that when severely deranged correlate with simple interventions that are expected to be available in the study settings and so provide information on unmet need. Multiple vital signs are used in compound scoring systems in some settings for identifying patients at risk. These systems require a consideration of several parameters that add complexity, and the summation into a score can be time-consuming and prone to error [9]. Furthermore, compound scores do not indicate the specific intervention that may be suitable for treating the patient.

The findings have particular relevance and urgency in the current global efforts to respond to the COVID-19 pandemic and to find strategies for implementing care with the greatest potential for positive impact. The global response to treatment of critical illness in COVID-19 has to-date mainly focused on advanced care such as mechanical ventilation with the World Bank alone providing 160 billion dollars of funding to the fight against COVID-19 [10].

The critically ill patients in the study were much younger than those cared for in ICUs globally [11] and many died in-hospital. High mortality rates among critically ill patients have been reported in Malawi and similarly in other low-income countries [12–14]. It may be that the unmet need of essential treatments is one explanation for high mortality rates. Reducing the unmet need (increasing the coverage of essential treatments), especially in general hospital wards where the majority of critically ill patients are cared for, may improve outcomes. An evaluation of unmet need may be useful as an indicator of quality and used in quality improvement interventions such as task-sharing of the initiation and modification of essential treatments to nurses and clinical officers when there are few physicians [4].

The large unmet need that we identified could be explained by several factors. It could be a sign of the general quality-of-care gaps that have been found to have a large impact on global mortality [15]. A lack of human resources causes a huge challenge for the health workers to prioritise among their duties [16]. There may be a lack of focus on the patients' illness severity and the principles of critical care due to specialty silos that prioritise diagnosing underlying pathologies and providing definitive treatments [5]. Critical illness may not be identified at arrival to hospital or following deterioration in the wards, or there may be a failure to initiate care once a critically ill patient has been identified.

Our study was limited by the inclusion of patients from only two hospitals and on five single days. The hospitals differ in size, referral level and organisation. Generalising the findings to other hospitals in Malawi and other settings should be done with caution and the prevalence of critical illness and the coverage of essential treatments may vary due to seasonal differences, availability of human and other resources and random variation. For some patients, an essential treatment may have been deemed inappropriate, potentially overestimating the unmet need. Our pragmatic definition of unmet need is based on interventions the patients were supposed to receive and this approach may lead to some misclassifications: there are reports that such interventions may not always improve outcomes for all patients [17].

The findings suggest a quality-of-care gap that may have a significant negative effect on outcomes and suggest that targeting critical care efforts in the COVID-19 response and beyond at improving the quality of essential care may have a greater impact than the introduction of

expensive, complex interventions such as mechanical ventilation [6]. Further research is required to understand the determinants of the unmet need of essential treatments for critical illness, the most effective implementation strategies to increase coverage of these low-cost treatments and their impact on patient outcomes.

## Conclusion

There was a large unmet need of essential treatments for critical illness in two hospitals in Malawi.

## Supporting information

**S1 Table. Patient characteristics by hospital.**
(DOCX)

## Acknowledgments

Thank you to all nurses and clinicians who assisted in the data collection the facilities.

## Author Contributions

**Conceptualization:** Raphael Kazidule Kayambankadzanja, Carl Otto Schell, Samson Kwazizira Mndolo, Markus Castegren, Tim Baker.

**Data curation:** Raphael Kazidule Kayambankadzanja, Isaac Mbingwani, Markus Castegren, Tim Baker.

**Formal analysis:** Raphael Kazidule Kayambankadzanja, Carl Otto Schell, Tim Baker.

**Funding acquisition:** Tim Baker.

**Investigation:** Raphael Kazidule Kayambankadzanja, Isaac Mbingwani, Samson Kwazizira Mndolo, Tim Baker.

**Methodology:** Raphael Kazidule Kayambankadzanja, Carl Otto Schell, Markus Castegren, Tim Baker.

**Project administration:** Raphael Kazidule Kayambankadzanja, Carl Otto Schell.

**Resources:** Raphael Kazidule Kayambankadzanja, Carl Otto Schell.

**Writing – original draft:** Raphael Kazidule Kayambankadzanja, Carl Otto Schell, Tim Baker.

**Writing – review & editing:** Raphael Kazidule Kayambankadzanja, Carl Otto Schell, Isaac Mbingwani, Samson Kwazizira Mndolo, Markus Castegren, Tim Baker.

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
