## [Decision Letter · Decision Letter 0]

6 May 2021

PONE-D-21-08206

Unmet need of essential treatments for critical illness in Malawi

PLOS ONE

Dear Dr. Kayambankadzanja,

Thank you for submitting your manuscript to PLOS ONE. After careful consideration, we feel that it has merit but does not fully meet PLOS ONE’s publication criteria as it currently stands. Therefore, we invite you to submit a revised version of the manuscript that addresses the points raised during the review process.

We look forward to receiving your revised manuscript.

Kind regards,

Tai-Heng Chen, M.D.

Academic Editor

PLOS ONE

 "NO: The funders had no role in the study design, data collection and analysis, decision to publish, or preparation of the manuscript. "

5. In your Methods section, please provide additional information about the methodology used in the study (for example, on the methods used to measure vital signs, and on a priori sample size calculations). 

Reviewers' comments:

Reviewer's Responses to Questions

**Comments to the Author**

1. Is the manuscript technically sound, and do the data support the conclusions?

Reviewer #1: No

Reviewer #2: Yes

Reviewer #3: Partly

2. Has the statistical analysis been performed appropriately and rigorously? 

Reviewer #1: No

Reviewer #2: N/A

Reviewer #3: No

3. Have the authors made all data underlying the findings in their manuscript fully available?

Reviewer #1: No

Reviewer #2: Yes

Reviewer #3: No

4. Is the manuscript presented in an intelligible fashion and written in standard English?

Reviewer #1: Yes

Reviewer #2: Yes

Reviewer #3: Yes

5. Review Comments to the Author

Reviewer #1: In this study, observers gathered data from a convenience sample of 1135 patients. The level of training and uniformity of training of the observers, and equipment used for measurements, are not specified. The data come from two different hospitals but the distribution of these data and the characteristics of the sample are not specified. Out of these 1135, 45 were identified as having an SpO2 <90, 103 a systolic BP < 90, and 17 had a GCS <9. (It's not clear if more than one of these features were seen in some of the patients)The reason for the study was to assess "unmet need" and the reason for looking for this need appears to be for "evaluating the processes of care and to highlight areas for potential quality improvement". The characteristics of these unwell patients are mainly medical, but the characteristics (gender, admitting specialty, age) of the contributing population remains undefined. The outcome in the selected patients is poor, probably worse in the surgical patients (but we are unaware of the outcome of the other possible 970). We don't even know which hospital they came from, or why three collection days were chosen in one hospital and two in the other.

Clearly there are two elements here: diagnosis (not as easy as it might appear, as indicated by citation 3), and then treatment which incorporates many factors: staff provision, their training, facilities, prognosis. It's of interest that there was no "unmet" need in the ICU, for example.

In other countries, this process would be termed "audit": good enough to highlight a problem, but not enough to tell us what to do about it. I suspect that the authors (quite a few, for such a simple data set) could easily have predicted this outcome. There has been a missed opportunity: the question that should be asked next is "why not"? Is there a lack of vigilance, or understanding, or resource?

Were these data anonymised when they were collected? If so, why are the anonymous data not available? If not, how were the data collected and stored?

Reviewer #2: Dr Kayambankadzanja and colleagues report in the what appears to be a point prevalence study on the unmet need of basic interventions in unwell patients in 2 Hospitals in Malawi. The study is in my view very interesting, timely and of value to a broad readership. It highlights the lack of basic interventions in low income, developing world countries and its association with mortality. Sadly patients are commonly very young and extrapolating from this and other data, many lives could potentially be saved and by relatively simple means. The current Pandemic has at times led to the risk of quite basic interventions such as administration of oxygen at sufficient levels being jeopardised in highly developed healthcare systems too, exceptional in the "developed world" yet common place in the developing world.

The paper is well written and concise, maybe a bit too concise.

I have a few comments to make:

As pointed out above, this appears to be a point prevalence study - in the Methods section / Abstract the 2nd sentence "All in patients aged >18 years on single days..." should be changed to the more scientific definition of what type of study this was - point prevalence.

The authors should provide additional basic data such as how many beds the 2 Hospitals have and the ratio of ICU beds to total Hospital beds so that readers get an impression of how scares a resource, monitored and reasonably equipped beds throughout this and many other parts of the world.

It would also be helpful to have a slightly more in detail split of demographics and pathologies of studied groups beyond "surgical, medical and Obs & Gyne? Simple and important information would be % of Trauma patients, number of sepsis, malaria etc.

I appreciate that the authors tried to keep the monitored parameters as relevant and easily measurable/simple as possible, but the additional capture of heart rate, respiratory rate and Temp would have been doable and informative. Given the young age of the studied patients a systolic blood pressure cut off of 90 may be slightly liberal especially if patients had a normal heart rate and were not shocked. Young and fit individuals present frequently with low systolic BP - can the authors comment please.

If additional parameters were not captured, a rationale should be provided in the Methods and possibly also Discussion section of the paper - it would be helpful to mention how other countries and healthcare systems go about managing at risk inpatients - in the UK for example via NEWS and more recently NEWS 2 scoring, other European countries and Australia use EWS too, in the US this is a more novel concept.

In terms of treatment, what is the setup apart from number of ICU beds in the 2 Hospitals studied. Is oxygen readily available and lastly is there a cost factor which has to be considered as well and needs mentioning i.e. are certain interventions not provided unless patients and relatives can pay for them?

Minor comment: Some minor language editing may be helpful for example: First sentence Background in the Abstract: should read "during" rather than in the Covid Pandemic

Reviewer #3: The authors submit a short research letter about critical care needs in Malawi. This is an interesting and important topic. However, I have some concerns about the current version of the manuscript:

1. The abstract should be more specific. Please outline in the background of the abstract and the introduction which vital signs you are investigating as markers of critical care requirements. For readers not familiar with LMIC critical care, it may be worth stating that delivery of O2 or fluid administration represents higher level of care in many hospitals. In the methods you state "The cut-offs for severe vital

70 sign derangements were adopted from previous work in Tanzania and Sweden (4, 7)", please refer to this in the introduction to explain why you used these markers and cut-offs.

2. Were there any differences between the DGH and the large hospital with 4 critical care beds regarding the delivery of care? Would it be possible to determine from the data you have? As mentioned above, it would link to how care is delivered in both settings: Does oxygen therapy and fluid administration happen outside intensive care? Routinely or only if no ICU bed is available? Are patients routinely screened for hypoxia and low blood pressure? If so, how often? Is GCS the routine measure for low GCS? How often is it preformed?

3. In the discussion the authors need to be specific that they have analysed data from two very different hospitals in Malawi. As to whether the lack of care provision is generalisable to other LMIC settings remains unclear. In my experience, there is a very wide variation of the care provided depending on country, localisation (urban vs rural), setup of healthcare system (private versus public, nationwide versus regional governance). These limitations must be mentioned.

6. PLOS authors have the option to publish the peer review history of their article (what does this mean?). If published, this will include your full peer review and any attached files.

Reviewer #1: **Yes: **Gordon Drummond

Reviewer #2: No

Reviewer #3: No

---

## [Author Response · Author response to Decision Letter 0]

14 Jun 2021

EDITOR’S COMMENTS

Please ensure that your manuscript meets PLOS ONE's style requirements, including those for file naming 

Thank you. We have followed the guidelines

Please provide additional details regarding participant consent. In the ethics statement in the Methods and online submission information, please ensure that you have specified (1) whether consent was informed and (2) what type you obtained (for instance, written or verbal, and if verbal, how it was documented and witnessed). If your study included minors, state whether you obtained consent from parents or guardians. If the need for consent was waived by the ethics committee, please include this information 

Thank you. We have added a sentence which reads: 

“Written, informed consent was obtained from all participants. For those with altered mentation, the participant’s next-of-kin provided the informed consent.

Thank you for stating the following financial disclosure:

 "NO: The funders had no role in the study design, data collection and analysis, decision to publish, or preparation of the manuscript. "

a. Please clarify the sources of funding (financial or material support) for your study. List the grants or organizations that supported your study, including funding received from your institution.

d. If you did not receive any funding for this study, please state: “The authors received no specific funding for this work.”

Thank you very much. We have included this information in the cover letter. It reads 

“TB received grants for the study from the Swedish Research Council, Laerdal Foundation, the Association of Anaesthetists of Great Britain and Ireland, Martin Rinds Stiftelse and Life Support Foundation.

 We note that you have indicated that data from this study are available upon request. PLOS only allows data to be available upon request if there are legal or ethical restrictions on sharing data publicly. For information on unacceptable data access restrictions, please see 

b) If there are no restrictions, please upload the minimal anonymized data set necessary to replicate your study findings as either Supporting Information files or to a stable, public repository and provide us with the relevant URLs, DOIs, or accession numbers. Please see for guidelines on how to de-identify and prepare clinical data for publication. For a list of acceptable repositories, please see 

Thank you. We have added a statement in the cover letter. It reads 

“Due to the sensitive nature of the information about severely unwell patients, and the requirements of the College of Medicine Research and Ethics Committee, data from the study are not publicly available. Data can be requested by contacting the corresponding author raphkazidule@gmail.com including a motivation for the request for the data and a detailed study plan.” 

In your Methods section, please provide additional information about the methodology used in the study (for example, on the methods used to measure vital signs, and on a priori sample size calculations). 

Thank you. We have provided additional information to the methods section – please see the manuscript. 

Reviewer 1

The level of training and uniformity of training of the observers, and equipment used for measurements, are not specified. 

Thank you for this comment. We have added the information in the methods. It now reads 

“We used Omron M2 and Lifebox pulse oximeters to measure blood pressure and oxygen respectively. Consciousness was measured using the Glasgow Coma Scale (GCS). Patients were considered to have an altered mental status if they had a GCS score of less than 9. Data were collected by qualified nurses and senior nursing students following training that was conducted the day before to ensure a standardised methodology.”

The data come from two different hospitals but the distribution of these data and the characteristics of the sample are not specified. 

Thank you. 

We have added a supplementary table showing patient characteristics by hospital.

The reason for the study was to assess "unmet need" and the reason for looking for this need appears to be for "evaluating the processes of care and to highlight areas for potential quality improvement". The characteristics of these unwell patients are mainly medical, but the characteristics (gender, admitting specialty, age) of the contributing population remains undefined. 

Thank you. 

We have now added the characteristics of the contributing patients in the Supplementary Table. 

The outcome in the selected patients is poor, probably worse in the surgical patients (but we are unaware of the outcome of the other possible 970). We don't even know which hospital they came from, or why three collection days were chosen in one hospital and two in the other. 

Thank you - we have added this information in the Supplementary Table. 

Thank you. We have added it reads 

“CDH had three data collections to increase representation from the facility”

Clearly there are two elements here: diagnosis (not as easy as it might appear, as indicated by citation 3), and then treatment which incorporates many factors: staff provision, their training, facilities, prognosis. It's of interest that there was no "unmet" need in the ICU, for example.

In other countries, this process would be termed "audit": good enough to highlight a problem, but not enough to tell us what to do about it. I suspect that the authors (quite a few, for such a simple data set) could easily have predicted this outcome. There has been a missed opportunity: the question that should be asked next is "why not"? Is there a lack of vigilance, or understanding, or resource? 

Thank you. 

We agree that the “why not” is really interesting, and have discussed this in the discussion section. We are currently planning further studies looking exactly at this question. We do believe that highlighting the unmet need of simple treatments that are readily available is an interesting findings per se and can spur further the work in this crucial field. 

Were these data anonymised when they were collected? If so, why are the anonymous data not available? If not, how were the data collected and stored? 

Thank you. 

The data were not anonymized on collection. We kept patient’s names and other identifying information for follow-up of patient outcomes. Data were anonymized prior to analysis and data are reported only at group-level. This process was reviewed and approved by the College of Medicine Research and Ethics committee. 

We have a statement in the methods on how data was collected. It reads “Data on patient demographics, diagnoses and prescribed treatments were collected from the patients’ files and clinical data on the patients’ vital signs and ongoing therapies were collected by direct clinical observation at the time of inclusion.”

REVIEWER 2

Reviewer #2: Dr Kayambankadzanja and colleagues report in the what appears to be a point prevalence study on the unmet need of basic interventions in unwell patients in 2 Hospitals in Malawi. The study is in my view very interesting, timely and of value to a broad readership. It highlights the lack of basic interventions in low income, developing world countries and its association with mortality. Sadly patients are commonly very young and extrapolating from this and other data, many lives could potentially be saved and by relatively simple means. The current Pandemic has at times led to the risk of quite basic interventions such as administration of oxygen at sufficient levels being jeopardised in highly developed healthcare systems too, exceptional in the "developed world" yet common place in the developing world.

The paper is well written and concise, maybe a bit too concise. 

Thank you. 

I have a few comments to make:

As pointed out above, this appears to be a point prevalence study - in the Methods section / Abstract the 2nd sentence "All in patients aged >18 years on single days..." should be changed to the more scientific definition of what type of study this was - point prevalence. 

Thank you. 

We have modified. It now reads 

“We conducted a point prevalence cross-sectional study…”

The authors should provide additional basic data such as how many beds the 2 Hospitals have and the ratio of ICU beds to total Hospital beds so that readers get an impression of how scares a resource, monitored and reasonably equipped beds throughout this and many other parts of the world. 

Thank you. We have added the information. It now reads 

“QECH is a 1500 bedded large, referral hospital with four adult ICU beds that provide mechanical ventilation, vasopressors and close monitoring. CDH is a 300 bedded district hospital without an ICU.”

It would also be helpful to have a slightly more in detail split of demographics and pathologies of studied groups beyond "surgical, medical and Obs & Gyne? Simple and important information would be % of Trauma patients, number of sepsis, malaria etc. 

Thank you for your comment. 

We have not included information on diagnoses as the underlying concept is that critical illness can affect all patients, and the emphasis is on illness severity rather than diagnosis. We agree that more information on the patients may have been useful, but we did not have the resources to collect accurate diagnostic information. 

I appreciate that the authors tried to keep the monitored parameters as relevant and easily measurable/simple as possible, but the additional capture of heart rate, respiratory rate and Temp would have been doable and informative. Given the young age of the studied patients a systolic blood pressure cut off of 90 may be slightly liberal especially if patients had a normal heart rate and were not shocked. Young and fit individuals present frequently with low systolic BP - can the authors comment please. 

Thank you. We included only these three vital signs as they have the clearest direct relationship with simple interventions that are expected to be available in the study settings (hypoxia and oxygen; hypotension and IV fluids; low conscious level and airway manoeuvres). 

We used a cut-off of systolic blood pressure of less than 90 as it is simple, and previous studies have shown it to be a marker of critical illness and associated with poor outcomes in hospitals. (Baker et al., 2015; Bell et al, 2006)

If additional parameters were not captured, a rationale should be provided in the Methods and possibly also Discussion section of the paper - it would be helpful to mention how other countries and healthcare systems go about managing at risk inpatients - in the UK for example via NEWS and more recently NEWS 2 scoring, other European countries and Australia use EWS too, in the US this is a more novel concept. 

Thank you. We have added some text to the methods: 

“Additional parameters were not included as they were not considered to correlate easily to simple interventions.”

And to the discussion:

“We chose three single vital signs that when severely deranged correlate with simple interventions that are expected to be available in the study settings and so provide information on unmet need. Multiple vital signs are used in compound scoring systems in some settings for identifying patients at risk. These systems require a consideration of several parameters that add complexity, and the summation into a score can be time-consuming and prone to error (Friman et al 2019). Furthermore, compound scores do not indicate the specific intervention that may be suitable for treating the patient.”

In terms of treatment, what is the setup apart from number of ICU beds in the 2 Hospitals studied. Is oxygen readily available and lastly is there a cost factor which has to be considered as well and needs mentioning i.e. are certain interventions not provided unless patients and relatives can pay for them? 

Thank you. We have added a sentence in the methods. It reads 

“Both of these are public facilities that provide acute care free of charge to all patients. QECH is a referral centre for CDH and other district hospitals”

Both hospitals have oxygen, but its availability at the right time for the right patients is not known and could be one of the underlying factors in the findings in this study. 

Minor comment: Some minor language editing may be helpful for example: First sentence Background in the Abstract: should read "during" rather than in the Covid Pandemic Thank you. We have reviewed the language. 

Reviewer 3

Reviewer #3: The authors submit a short research letter about critical care needs in Malawi. This is an interesting and important topic. However, I have some concerns about the current version of the manuscript: 

Thank you.

1. The abstract should be more specific. Please outline in the background of the abstract and the introduction which vital signs you are investigating as markers of critical care requirements. For readers not familiar with LMIC critical care, it may be worth stating that delivery of O2 or fluid administration represents higher level of care in many hospitals. In the methods you state "The cut-offs for severe vital 70 sign derangements were adopted from previous work in Tanzania and Sweden (4, 7)", please refer to this in the introduction to explain why you used these markers and cut-offs. 

Thank you - we have amended the abstract and the text in the body of the manuscript.

2. Were there any differences between the DGH and the large hospital with 4 critical care beds regarding the delivery of care? Would it be possible to determine from the data you have? As mentioned above, it would link to how care is delivered in both settings: Does oxygen therapy and fluid administration happen outside intensive care? Routinely or only if no ICU bed is available? Are patients routinely screened for hypoxia and low blood pressure? If so, how often? Is GCS the routine measure for low GCS? How often is it preformed? 

Thank you for this good comment. 

We have added a supplementary table that shows the results by hospital. Yes, oxygen and fluids are intended to be given routinely outside ICU in both hospitals. We do not have information about routine or triage screening in the hospitals – a lack of systematic identification of critical illness could be an underlying reason for the findings of large unmet needs in the study. We have added a statement describing that these treatments are provided in the general wards. 

“The treatments were chosen as they are regarded as standard medical practice and were expected to be available in all wards in both the study hospitals” 

3. In the discussion the authors need to be specific that they have analysed data from two very different hospitals in Malawi. As to whether the lack of care provision is generalisable to other LMIC settings remains unclear. In my experience, there is a very wide variation of the care provided depending on country, localisation (urban vs rural), setup of healthcare system (private versus public, nationwide versus regional governance). These limitations must be mentioned 

Thank you. We have amended the discussion.

---

## [Decision Letter · Decision Letter 1]

5 Aug 2021

Unmet need of essential treatments for critical illness in Malawi

PONE-D-21-08206R1

Dear Dr. Kayambankadzanja,

We’re pleased to inform you that your manuscript has been judged scientifically suitable for publication and will be formally accepted for publication once it meets all outstanding technical requirements.

Kind regards,

Tai-Heng Chen, M.D.

Academic Editor

PLOS ONE

Additional Editor Comments (optional):

Please make corrections according to Reviewer 3's comments. Afterwards, this manuscript would be completely acceptable.

Reviewers' comments:

Reviewer's Responses to Questions

**Comments to the Author**

1. If the authors have adequately addressed your comments raised in a previous round of review and you feel that this manuscript is now acceptable for publication, you may indicate that here to bypass the “Comments to the Author” section, enter your conflict of interest statement in the “Confidential to Editor” section, and submit your "Accept" recommendation.

Reviewer #1: All comments have been addressed

Reviewer #2: All comments have been addressed

Reviewer #3: All comments have been addressed

2. Is the manuscript technically sound, and do the data support the conclusions?

Reviewer #1: (No Response)

Reviewer #2: Yes

Reviewer #3: Yes

3. Has the statistical analysis been performed appropriately and rigorously? 

Reviewer #1: (No Response)

Reviewer #2: N/A

Reviewer #3: No

4. Have the authors made all data underlying the findings in their manuscript fully available?

Reviewer #1: (No Response)

Reviewer #2: Yes

Reviewer #3: No

5. Is the manuscript presented in an intelligible fashion and written in standard English?

Reviewer #1: (No Response)

Reviewer #2: Yes

Reviewer #3: Yes

6. Review Comments to the Author

Reviewer #1: (No Response)

Reviewer #2: (No Response)

Reviewer #3: Thank you for addressing all comments. It would be nice to see some basic statistics (e. g. Chi square) to compare hospitals. Also, please delete "Or" in line 142. Start the sentence with "There...."

7. PLOS authors have the option to publish the peer review history of their article (what does this mean?). If published, this will include your full peer review and any attached files.

Reviewer #1: No

Reviewer #2: No

Reviewer #3: No

---

## [Editor Report · Acceptance letter]

2 Sep 2021

PONE-D-21-08206R1 

Unmet need of essential treatments for critical illness in Malawi 

Dear Dr. Kayambankadzanja:

I'm pleased to inform you that your manuscript has been deemed suitable for publication in PLOS ONE. Congratulations! Your manuscript is now with our production department. 

Kind regards, 

on behalf of

Dr. Tai-Heng Chen 

Academic Editor

PLOS ONE